# Development of a Reverse Logistics Modeling for End-of-Life Lithium-Ion Batteries and Its Impact on Recycling Viability—A Case Study to Support End-of-Life Electric Vehicle Battery Strategy in Canada

**Giovanna Gonzales-Calienes** * , **Ben Yu and Farid Bensebaa**

Energy, Mining and Environment Research Centre, National Research Council Canada, 1200 Montreal Road, Ottawa, ON K1A 0R6, Canada

* Correspondence: giovanna.gonzalescalienes@nrc-cnrc.gc.ca; Tel.: +1-613-9931700

**Abstract:** The deployment of a sustainable recycling network for electric vehicle batteries requires the development of an infrastructure to collect and deliver batteries to several locations from which they can be transported to companies for repurposing or recycling. This infrastructure is still not yet developed in North America, and consequently, spent electric vehicle batteries in Canada are dispersed throughout the country. The purpose of this reverse logistics study is to develop a spatial modeling framework to identify the optimal locations of battery pack dismantling hubs and recycling processing facilities in Canada and quantify the environmental and economic impacts of the supporting infrastructure network for electric vehicle lithium-ion battery end-of-life management. The model integrates the geographic information system, material flow analysis for estimating the availability of spent battery stocks, and the life cycle assessment approach to assess the environmental impact. To minimize the costs and greenhouse gas emission intensity, three regional recycling clusters, including dismantling hubs, recycling processing, and scrap metal smelting facilities, were identified. These three clusters will have the capacity to satisfy the annual flow of disposed batteries. The Quebec–Maritimes cluster presents the lowest payload distance, life-cycle carbon footprint, and truck transportation costs than the Ontario and British Columbia–Prairies clusters. Access to end-of-life batteries not only makes the battery supply chain circular, but also provides incentives for establishing recycling facilities. The average costs and carbon intensity of recycled cathode raw materials are CAD 1.29/kg of the spent battery pack and 0.7 kg $CO_{2e}$/kg of the spent battery pack, respectively, which were estimated based on the optimization of the transportation distances.

**Keywords:** end-of-life lithium-ion battery; battery recycling; geographic information system; material flow analysis; GHG emissions; transportation network optimization; reverse logistics

## 1. Introduction

Greenhouse gas (GHG) emission reduction is a crucial target to fight climate change, and accelerating decarbonization in diverse economic sectors is a net-zero emissions pathway to achieve it. Several national governments worldwide have announced net-zero GHG emission pledges (in law, proposed, and policy documents), which commit to developing long-term, low GHG emission strategies in line with limiting temperature increases to 1.5 °C above preindustrial levels by 2050. Traditionally, transportation has been heavily reliant on fossil fuels, which accounted for more than 90% of the transport sector energy needs in 2020. Electrification plays a central role in decarbonizing light- and heavy-duty vehicles, which rely on policies to promote electric mobility (battery- or fuel cell-powered electric vehicles) [1].

The electrification of road transportation entails challenges related to battery supply chain sustainability and end-of-life (EoL) management. Technology advancements in

lithium-ion batteries (LIBs) have become the main drivers for accelerating the development of the electric vehicle (EV) market. The raw material inputs associated with LIBs, such as lithium, nickel, cobalt, and graphite, present a high level of criticality with the associated high costs and environmental impacts [2–4]. EoL management of spent EV LIBs is an important aspect of the closed-loop life cycle approach for EV LIBs. Reusing and recycling LIBs will reduce the dependence on mining and refining of the critical materials in LIBs, decrease the negative environmental impact, and generate local socioeconomic benefits. Before the repurposing and recycling phases of EV LIBs, LIBs must be collected and dismantled, and key valuable components must be transported to repurposing or recycling facilities.

The transportation of spent EV LIBs is a critical activity that influences the environmental impact and costs of EoL management. A current review of state-of-art of spent EV LIBs transportation emphasizes the importance of the strategic design of a reverse logistics network of spent EV LIBs [5–7]. Wilson and Goffnett [8] describe the reverse logistics in supply chain management as the flow of materials in the reverse direction to the main flow. In the case of spent batteries, reverse logistics deal with the collection of EoL LIBs from the end users and the transport of such batteries to downstream facilities including remanufacturing, reusing, and recycling [5]. Some studies suggest that optimized facility siting to minimize distances decreases transportation costs and emissions [9–13]. Additionally, it is implied that a reduction in transportation distances with a design of local dismantling facilities also reduces safety risks [10,14,15] and reduces unnecessary transportation costs when spent batteries are sent to recycling facilities and the remaining can be reused [14,15].

There is limited research on optimization research on EoL infrastructure siting and the role of transportation of spent batteries for EoL processing stages. This research is critical to evaluate the viability of EoL recycling, such as the optimal locations of EoL management facilities, while assessing the environmental and cost impacts of collection and transportation. The recycling viability of building EoL management facilities should be evaluated with a view to different factors, such as recycled material amounts and prices, transport distances, financial metrics, policies, and types of technology [16].

Table 1 provides key information on the relevant studies in this field. Some studies on EV LIBs reverse logistics have been focused on modeling the optimization of siting locations of EoL facilities. Wang et al. [13] designed an optimal recycling network in China assuming transportation costs and carbon taxes. Hoyer, et al. [17] developed an optimization model for technology and capacity planning in recycling networks, where the strategy of setting up all the recycling plants at the beginning of the planning horizon presented the higher net present value. Tadaros, et al. [18] developed a reverse supply chain model for used LIBs in Sweden. The model is defined as an unlinked discrete multiperiod problem without taking into account any inventories. Hendrickson, et al. [19] used geographic information system (GIS) and life cycle assessment (LCA) to assess the environmental impacts of used battery supply logistics beyond GHG emissions, highlighting the impact of transportation on human health damages in California and pointing out the importance of an optimized system design for siting new facilities. Although Rallo, et al. [20] did not use an optimization modeling approach, their research work analyzed the economic viability of centralized and decentralized dismantling scenarios in Europe and concluded that a decentralized facility scenario can reduce logistics costs and $CO_{2e}$ emissions rather than a centralized configuration of dismantling facilities. Nguyen-Tien, et al. [21] investigated the impact of travel distance of transporting spent batteries on the business and economic viability of building EoL recycling facilities.

**Table 1.** Studies that report a reverse logistics optimization for spent EV LIBs and the environmental and economic impacts of transporting spent batteries. Adapted from [5,7].

| Studies | EoL Stage | Modeling Approach | Decision Variable | Spent Battery Transportation Impact | Geographical Scope | Assumptions |
|---------|-----------|-------------------|-------------------|-------------------------------------|--------------------|-------------|
| [13] | Collection, recycling, disposal | Mixed-integer linear programing | Optimal facility location | Economic and environmental | China | Travel distances are estimated as straight-line distances between collection and recycling facilities. Baseline emissions not reported. |
| [17] | Collection, recycling | Mixed-integer linear programing | Optimal investment plan | Economic | Germany | Transportation cost decreases in a decentralized collection facility design. |
| [18] | Inspection, recycling | Mixed-integer linear programing | Optimal facility location and allocation of demand zones to each facility | Economic | Sweden | Assumes different transport modes to optimize transportation cost. |
| [19] | Collection, dismantling, recycling | GIS and LCA | Optimal facility location | Environmental | California | Transportation cost value is not specified. |
| [20] | Dismantling | Economic and environmental assessment | - | Economic and environmental | Europe | Germany as centralized scenario and Spain as decentralized network |
| [21] | Collection recycling | Material flow analysis Geospatial supply chain model Economic and environmental assessment | Optimal facility location | Economic and environmental | UK | Recycling demand distributed equally between collection sites. Transportation costs assumed from EverBatt model [22,23] |

With few peer-reviewed publications on the optimization of reverse logistics for spent EV LIBs, our work aims at addressing three relevant gaps of the current state of reverse logistics research as highlighted above, i.e., the forecasted demand for LIB recycling, optimization of EoL management facilities, and economic and environmental assessments. For that purpose, this study developed a detailed spatial model for reverse logistics in Canada by integrating material flow analysis, geographical information system tools, transportation costs, and life cycle approach for GHG emissions. The contribution of this study is to include new spatial modeling parameters and criteria for spent battery supply chains as follows: (i) local demographical, geographical, and socioeconomic factors to geolocate future spent battery collection sites and allocate recycling demand, (ii) siting location criteria and routing optimization procedures applied to regional recycling clusters for future collection sites, dismantling hubs, and recycling facilities, and (iii) recovering materials from the whole battery pack, i.e., the battery cell and the balance-of-system components.

With the fast growth of LIB demand, EV LIB recycling will play a critical role in the sustainability of the electrification of the transport sector in Canada and elsewhere. The 2030 Emissions Reduction Plan presents the goal to reach a new climate target of cutting emissions by 40% below 2005 levels by 2030 in order to achieve net-zero emissions by 2050. Transportation accounted for 25% of total GHG emissions in 2019, and its carbon footprint has risen 16% in the last 17 years. A key milestone is to reach 100% zero-emission electric vehicles (ZEVs) in total light-duty vehicle (passenger cars and light commercial vehicles) sales by 2035 [24]. A ZEV is considered a battery electric vehicle (BEV), plug-in hybrid electric vehicle (PHEV), and fuel cell electric vehicle (FCV). With no commercial deployment for FCVs, BEVs and PHEVs will continue to dominate the ZEV market, and it is expected that it will continue to grow in order to reach Canada's net-zero targets.

Hence, this study aims to understand how the transportation of spent EV LIBs has an influence on the recycling viability in Canada. The purpose of this study is to develop a reverse logistics modeling framework for EoL lithium-ion batteries, combining Geographical Information System (GIS), Material Flow Analysis (MFA), and Life Cycle Assessment (LCA) tools and propose a case study, which provides for the first time a detailed framework

for recycling clusters of spent EV LIBs in Canada. The case study proposes an optimized transportation network that minimizes transportation distances to identify the best siting locations for dismantling hubs and recycling facilities and allocates battery mass for recycling while evaluating the associated costs and environmental impacts. Furthermore, the overall life-cycle GHG emissions and costs of EV LIBs at the EoL phase are completed by including life-cycle GHG emissions and the associated costs of transporting spent batteries along the reverse logistics network system. The overall life-cycle GHG emissions of recycling processes to recover EV LIB cathode materials can be compared with the EV LIB cathode production from virgin materials.

## 2. Methods

A spatial model of a transportation network for spent batteries was developed to identify and optimize siting of dismantling hubs and recycling processing facilities in Canada to minimize the costs and environmental impacts of spent battery transportation by using GIS.

This spatial modeling was integrated with MFA and LCA methods and estimated the quantity of spent EV LIBs viable for recycling by 2040, the transportation distance, and the potential GHG emissions and costs related to transportation of spent batteries from collection centers to EoL processing facilities.

In this study, we focused on the reverse logistics of battery recycling clusters. Figure 1 depicts the overall reverse logistics network considered for one recycling cluster. This included three types of actors or activities: collection, dismantling, and recycling.

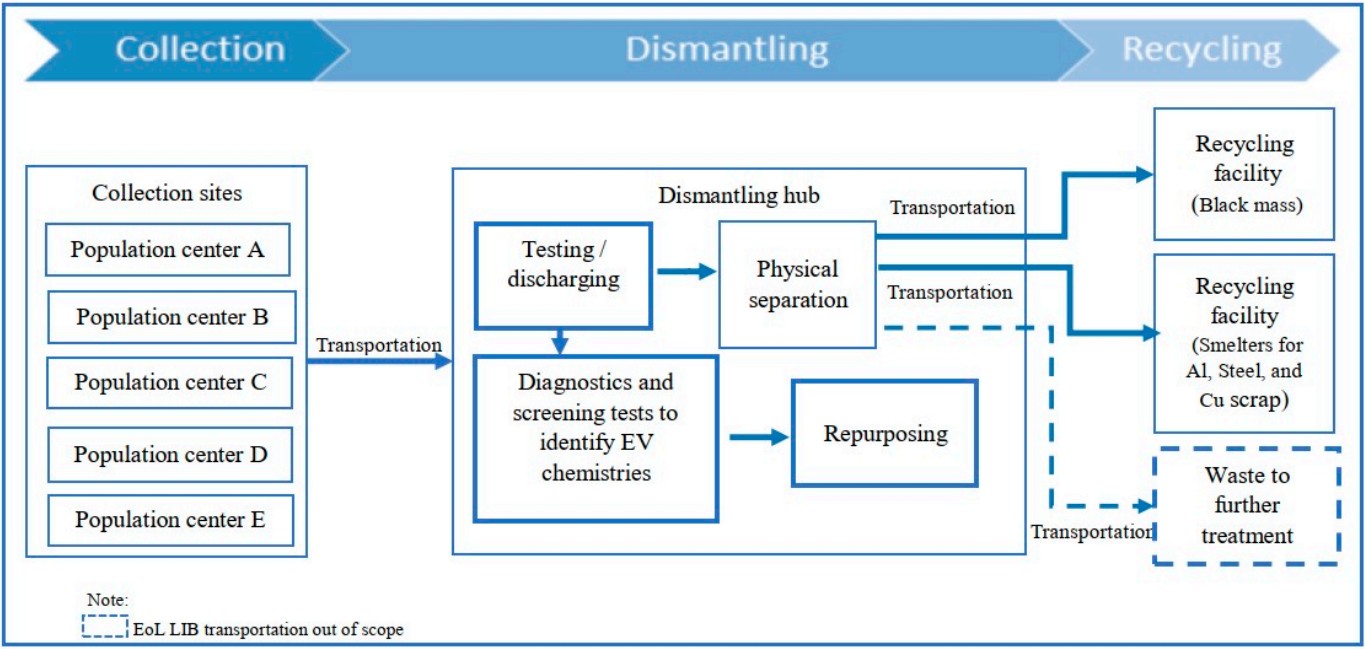

**Figure 1.** High-level representation of spent EV LIB packs in reverse logistics network in a recycling cluster, which has three main EoL management facilities, predefined collection sites located in major population centers. Recycling demand is allocated to each facility then spent batteries are transported to dismantling hubs to be diverted to repurposing and recycling processes. The battery pack is separated into black mass (battery cell materials) and other metals before being sent to recycling facilities.

End-of-life EV LIBs are collected and transported to dismantling hubs for discharging and testing processes to assess suitability for either repurposing or recycling [25]. Spent EV LIBs are transported to a recycling facility after physical separation (dismantling). Our model did not include the transportation of batteries from EV users to the collection points, which is a common practice in the literature [26].

Because only battery cells should be shipped to a centralized battery recycling facility for raw materials recovery [19], the main pathways identified for the spent battery stream (Figure 1) started with collection of these spent LIB packs that are transported to dismantling hubs to be physically separated by shredding. Second-life applications, such as repurposing of end-of-life batteries, is an optional pathway that would potentially modify the distribution of the batteries before recycling, and this would have implications on the facility optimization process described in this study. Hence, the repurposing pathway was considered in the current study by estimating an annual percentage of the total EV battery packs to be repurposed.

After dismantling the battery pack, the battery cell components, such as anode, cathode, carbon black, and binder, are crushed to produce a black mass (filter cake). The cathode chemistry will affect the relative Li, Ni, Mn, Co, and graphite contents of this black mass. Plastics and electrolytes from the battery cell are sent to further waste treatments. The battery cell's nonhazardous components, such as aluminum from the cathode current collector, positive terminal assembly, and cell container and copper from the anode current collector and negative terminal assembly, may be separated and transported to smelters depending on whether pyrometallurgical or hydrometallurgical recycling processes are used. In the case of a pyrometallurgical process, the nickel and cobalt in the black mass and copper can be recovered, but aluminum in the black mass is lost. In the case of a hydrometallurgical process, the black mass does not contain aluminum and copper. The final destinations of the battery cells (as black mass) are centralized recycling processing facilities to recover metal sulfates (pyrometallurgical process) or cathode materials (hydrometallurgical process) to be reused in cathode and battery cell production, respectively.

The balance-of-system components (BOS), often referred to as battery management systems, module and battery pack terminals, heat conductors, and module and pack enclosures, are separated from the battery cell and are also dismantled, sorted, and shredded to separate metals, such as aluminum, copper, and steel scraps. These metals are sold and transported by truck to a few dedicated smelters across Canada that process them as secondary material inputs. Transportation to smelting facilities was included in the scope of this study. However, other BOS components, such as plastics, electronic scraps, used cables, and used printed wiring boards, were considered waste materials and sent to waste treatment facilities. The transportation of these waste materials for further treatment was out of the scope of this study.

*2.1. Spatial Modeling*

Building on reverse logistics studies [15,21,23], a spatial modeling using GIS network analysis software was developed. This model enabled the development of an efficient transportation network of spent batteries along their supply chain including collection sites through dismantling hubs and recycling processing facilities. Transportation costs and GHG emissions can be reduced by identifying the shortest path between several origins and destinations. Network optimization in this study was performed using ArcGIS Pro© 3.0.1, software developed by Esri Inc., Toronto, ON, Canada.

Initially, a modeling framework combining GIS, MFA, and LCA was developed, and it is schematically summarized in Figure 2. The GIS optimization of dismantling hubs and recycling locations comprised four steps. First, network datasets were created; second, origin and destination sites were identified; third, the mapping of different sites was established; and finally, location-allocation was optimized [19,27]. Once the four steps were completed, optimized truck transportation distances from the collection sites to the recycling processing facilities were applied to the life-cycle GHG emission intensity of trucks on road transportation networks. This was done to obtain life-cycle environmental emissions related to the transportation of spent EV batteries, which complemented the life-cycle emission values of the battery recycling processing. Additionally, route heat maps were generated to highlight the hot spots of route transportation of each reverse logistics segment.

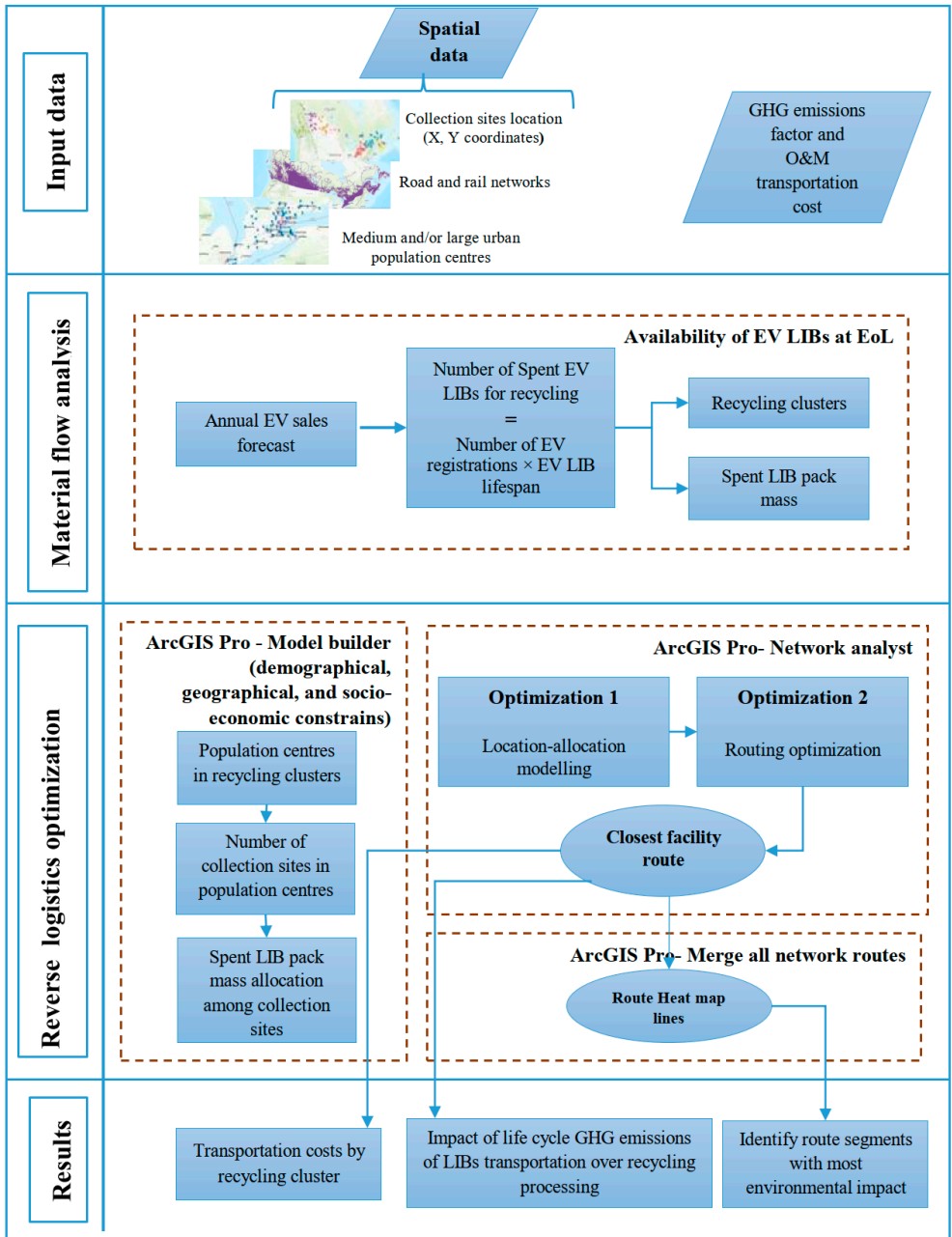

**Figure 2.** Proposed assessment framework combining GIS, MFA, and LCA tools to model transportation of spent EV batteries in a recycling cluster. This framework comprises three modules starting with input data (geospatial data, parameters and assumptions, and environmental and economic factors) follow by an MFA module to estimate recycling demand. Input data and MFA modules feed the reverse logistics optimization module to optimize EoL facility locations. Results: Module analyzed the economic and environmental impacts on recycling viability.

For this GIS-based spatial modeling approach, some assumptions were made to optimize locations and capacities of facilities, which directly affected truck transportation distances. The following GIS model criteria and constraints were listed for this study: (i) Availability of spent EV LIBs for EoL management by Canadian provinces, which depends on the annual demand forecast of EV LIBs packs; (ii) Allocation of available spent battery mass among collection sites in urban population centers. Distribution of the spent battery pack supplies among different collection sites may depend on demographic, geographic, and socioeconomic constraints; (iii) Black mass available for the recycling process-

ing is based on average battery mass components contribution, which depends on battery chemistry; (iv) Potential number of dismantling and recycling facilities considers a flexible plant capacity with an estimated minimum and maximum plant capacity throughput based on its economic feasibility; (v) Placement of dismantling and recycling facilities to minimize the environmental impacts from on-site emissions should consider economic impacts, such as (a) access to a qualified labor force and (b) end point utilization with access to critical infrastructure, such as rail transportation systems, that may facilitate transportation to battery production facilities; (vi) GIS network analysis depends on the transportation travel type and mode (single/intermodal) and the location–allocation problem type under the network analysis tool.

Changes in criteria of facility placement are expected to have a significant impact on travel distances [19]. Consequently, this will affect GHG emissions and transportation costs.

### 2.1.1. Availability of Electric Vehicle LIBs at EoL

The spatial transportation model criteria were integrated to a material flow analysis (MFA) to estimate the number of spent EV LIBs that were available for collection. The MFA methodology used in the Moore, Russell, Babbitt, Tomaszewski and Clark [27] study, which used a market lifespan approach [27,28], was selected to be applied in this study. LIBs reaching their EoL were estimated based on past registrations of new EVs and average values of EV LIB life span (Equation (1)).

$$S_{B,t} = B_{(t-l)} \times P_l \tag{1}$$

$S_{B,t}$ represents the number of spent EV LIBs that are reaching EoL and are available for collection in year $t$. $B_{(t-l)}$ reflects the number of LIB packs starting useful life in EVs registered in the past year $(t - l)$ based on the expected LIB lifespan in years ($l$), and $P_l$ reflects the percentage of EV LIBs that will reach their EoL after $l$ year of lifespan. It was assumed that each EV includes a single LIB pack, and EoL options are recycling and repurposing (reuse) with an estimated share of 70% and 30%, respectively, over the total waste stream in year $t$. [14,29]. EV average lifespan distribution data ($l$ and $P_l$) were adapted from Richa, et al. [30]. This study considered four lifespans of 6, 8, 10 and 15 years.

In this study, the collection of spent EV Li-ion batteries was limited to zero-emission electric vehicles (ZEVs) ($S_{B,t}$), including full battery electric vehicles (BEVs) and plug-in hybrid electric vehicles (PHEVs) as defined by Transport Canada [31]. To estimate the quantity of batteries placed on the market each year, the historical number of new ZEVs registered in Canada from 2011 to 2021 by Statistics Canada [32] and the ZEV forecast from 2022 to 2040 [33] were considered. On 1 May 2019, the Government of Canada launched the national rebate program as part of the Incentive for Zero-Emission Vehicles, offering a rebate of up to CAD 5,000 with the purchase of a zero-emission vehicle. ZEV reached a 5.2% share of total light-duty vehicle registrations in 2021 [34]. In addition, BEVs and PHEVs represented 68% and 32%, respectively, of the total new ZEV registrations in 2021. Table S1 in the Supplementary Information shows the annual new ZEV registrations from 2011 to 2021.

In Canada, the new 2030 Emissions Reduction Plan presents the goal to reach a new climate target of cutting emissions by 40% below 2005 levels by 2030 in order to achieve net-zero emissions by 2050 [24]. Since transportation accounted for 25% of total GHG emissions in 2019, and its carbon footprint has risen 16% in the last two decades, a 100% ZEV sales target by 2035 for light-duty vehicle (passenger cars and light commercial vehicles) was set [24].

The ZEV registrations forecast for 2040 can consider two scenarios, a baseline scenario and a net-zero target scenario. Sections S2 and S3 in the Supplementary Information describe the assumptions for both forecast scenarios, and Tables S2 and S3 summarize the EV LIB inflow and waste streams for recycling for the baseline and net-zero MFA scenarios, respectively.

Annual new electric vehicle registrations in Canada are not the same across all provinces [34]. In 2021, Quebec, Ontario, and British Columbia contributed the most registrations at 43%, 23%, and 28%, respectively, of total ZEV sales. Note that the ZEV incentive program, which is a key driver of EV adoption, is a mandate in British Columbia and Quebec as there is an Internal Combustion Engine (ICE) ban by 2030 and 2040, respectively. However, Ontario, the largest province by population, has no ZEV mandate, and there is no ICE ban [35].

Battery pack availability across Canada at the provincial level by 2040 was estimated by assuming the market share of the number of ZEVs by province in 2040 would be the same as the market share in 2021 [34]. Based on this estimated spent battery distribution by province, this study assumed three geospatial centralized recycling clusters to place dismantling hubs and recycling processing facilities in Canada with the capacity to satisfy the annual disposed battery flows in Canada. One was located in the West (British Columbia) covering spent batteries from four provinces (BC and the Prairies provinces AB, MB, and SK), one was located in Ontario (covering Ontario), and one was located in Quebec (covering QC and the Maritimes provinces NB and NS). The centralized recycling cluster scenarios assumed that batteries are transported to processing facilities from different provinces to allow for larger centralized facilities that would benefit from economies of scale and easier access to batteries and markets for recovered materials beyond a specific province. Tables S4 and S5 indicate the battery mass allocation among provinces for a baseline scenario and a net-zero target scenario, respectively. For this study, the spent battery mass allocation baseline scenario was used for the reverse logistics optimization case study developed in the following sections.

### 2.1.2. Reverse Logistics Optimization

To model the reverse logistics, an optimization analysis was used to estimate the optimal number of dismantling hubs and recycling processing facilities and their locations in the road transportation networks of the West and East recycling clusters. The objective function of location-allocation optimization was to minimize the transportation distances, expressed as total ton–kilometers transported, between the collection sites and the intermediate and final EoL processing destinations.

### Allocation of Available Spent Battery Mass among Collection Sites in Population Centers

Some spent batteries may come through official original equipment manufacturer (OEM) channels and other car dealers, given the need for trained mechanics to service EVs. However, car dealerships are not required to take back spent EV batteries for disposal. Rather, most EV batteries at EoL will come through the auto dismantling or auto recycling supply chain [36]. In Canada, the Automotive Recyclers of Canada Association (ARC) represents approximately 400 end-of-life vehicle recycling and dismantling facilities throughout Canada. Among them, some typical vehicle scrapyard facilities have started to collect an increasing number of spent EV batteries. At that point, scrapyard staff at these facilities have been receiving training on EV battery EoL management, including diagnostics and safety discharging of EV batteries [36]. This study assumed that spent EV battery collection facilities would be confined to pre-existing EV scrapyards across Canada, which can provide the infrastructure to store, dismantle from EVs, and handle EV batteries properly. A registry of automotive scrapyards exists online through the website of the Automotive Recyclers of Canada Association [37]. The geospatial data layer of EV scrapyards is illustrated in Figure S1 in the Supplementary Information and shows a map of the geolocations of spent EV battery collection sites across Canada.

The disposed EV battery pack mass per provincial cluster for the baseline scenario was estimated in Section 2.1.1 and is shown in Table S4 in the Supplementary Information and is allocated among the different collection sites located in population centers (PCs) across Canada, to which geospatial data are provided by Statistics Canada [38,39] and illustrated in Figures S2 and S3 in the Supplementary Information. In order to identify strategic

geolocations for collection sites to decrease payload distances, demographic, geographic, and socioeconomic constraints were applied by using geoprocessing tools. Collection sites would be located within 30 km outside the borders of medium and large urban PCs, whose population size is more than 30,000 based on population counts from the 2016 census [40], and applied a weighted allocation of the spent battery pack mass based on household income data. Using ModelBuilder of ArcGIS Pro© 3.0.1, software developed by Esri Inc., Canada, a geoprocessing model was built to automate workflows that string together sequences of geoprocessing tools, feeding the output of one tool into another tool as input [41]. A description of the workflow for the allocation of battery mass among collection sites is presented in Section S7 in the Supplementary Information.

Location-Allocation of Dismantling Hubs and Recycling Processing Facilities and Routing Optimization

To optimize the closest facility siting, transportation network datasets were set up using ArcGIS Pro© software environment. Building geospatial road transportation networks and other required geospatial data, such as locations of major cities, rail stations, and borders of cities, were sourced from transport networks as digital cartographic reference products produced by Natural Resources Canada [42,43] and Statistics Canada [44].

The locations of the dismantling and recycling facility candidates were assumed to be industrial zones located within the medium and large urban population centers, serving as a useful approximation for the purposes of this study [45]. Table S6 in the Supplementary Information shows the potential dismantling and recycling facility location candidates. An important feature of the model is that it allows the user to limit the number of facility candidates to those that fulfill certain siting criteria, such as proximity to large PCs and rail infrastructure. At this stage, the entire spent EV battery packs are transported to dismantling hub facilities where the black mass, i.e., battery cell components (cathode/anode materials), is produced and assumed to be 41% of the total battery pack mass allocated to each dismantling facility chosen. The black mass is delivered to either traditional recycling processing facilities to recover Co, Ni, and Cu (pyrometallurgical plants) or Li, Ni, Co, and Mn (hydrometallurgical plants). Plastics and electrolytes from battery cells and solvents and electronic scrap from BOS represent 20% of the battery pack mass and are delivered to a waste treatment facility. The remaining 39% of the battery pack mass from the battery cell and balance of system containing nonhazardous materials (Cu, Al, and steel scraps) are sent to recycling processing using smelters, where copper, aluminum, and steel scraps represent 18%, 19% and 2%, respectively, of the total battery pack mass allocated to each dismantling facility chosen. NMC811 was assumed to be the only EV LIB chemistry in the spent battery stream because it is one of the predominant battery chemistries in the current EV market [46]. Mass contribution of the EV LIB pack was based on Dai, Kelly, Gaines and Wang [22]'s study, which presented detailed material compositions at the cell, module, and pack levels.

We assumed that the actual smelters used as recycling processing facilities to process aluminum, copper, and steel scraps were located in the West and East clusters. Table S7 in the Supplementary Information shows the aluminum, steel, and copper smelting facility location candidates.

To identify the best locations for dismantling and recycling processing facilities, an algorithm was developed for the geospatial optimization of the dismantling and recycling facility sites using a location-allocation methodology that integrated the economic and environmental metrics into the segments of a GIS network [19,47]. The location-allocation optimization process was applied by using the ArcGIS Network analysis tool in ArcGIS Pro© software, in which locations of the collection sites represented where the demand point locations and potential dismantling sites were considered as facilities. For the other GIS segments, the dismantling facilities chosen were the demand points, and the recycling and smelter locations were the facilities. This optimization problem that is to minimize weighted impedance (P-Median) was solved by the location-allocation solver. Given N

candidate facilities and M demand points with a weight, we chose a subset of the facilities, P, such that the sum of the weighted distances from each M to the closest P was minimized based on the Dijkstra's algorithm [48]. The values of the fields' demand weight and impedance corresponded to the allocated battery mass to each demand point and travel distance between the demand point and facility chosen, respectively. The allocated battery mass for the recycling and smelting facilities was calculated using geoprocessing tools in ArcGIS Pro©.

Routing optimization was used to generate route stops from location-allocation outputs. This network analysis type in ArcGIS Pro© optimized and merged the routes of the overlapping three GIS segments into one and summed their weight, i.e., collection sites to dismantling hubs chosen, dismantling sites chosen to recycling processing facilities chosen, and dismantling sites chosen to smelters, by using the ModelBuilder and the Join, Merge, Buffer, Intersect, and Dissolve geoprocessing tools [49].

### 2.1.3. Life-Cycle GHG Emissions and Transportation Costs

Utilizing a life cycle perspective in evaluating the overall spent EV battery transportation from collection sites through dismantling hubs to a centralized recycling facility is essential in fully understanding the environmental consequences of this infrastructure expansion. An LCA approach was used in this study to estimate the life-cycle GHG emissions of the reverse logistics of EV LIBs. A transportation system boundary comprised collection sites to recycling processing facilities (battery cell and other battery pack metal recycling processing). In order to integrate transportation LCA results with the life cycle emissions from spent EV battery pack recycling processing, an average life-cycle GHG emissions for recycling processing was assumed as proxy for each recycling cluster with a value of 0.678 $kgCO_{2e}$/kg of the spent battery pack [50–52]. The assumptions to estimate the GHG emission intensities and the truck transportation unit costs used in this study are explained in Section S10 in the Supplementary Information.

### 3. Results and Discussion

*3.1. Location-Allocation of EoL Processing Facilities in Recycling Clusters and Route Optimization for Transportation of Spent EV LIBs to Recycling Facilities*

The first step on the spatial analysis model in this study was to identify the suitable collection sites in population centers for each recycling cluster in Canada based on siting criteria, which included close distance to population centers with a population of more than 50,000. In addition to the criteria to select the recycling clusters explained in Section 2.1.1, these three recycling cluster scenarios were selected based on the following: (i) Three major cluster regions including Ontario or ON covering the center of Canada, Quebec (QC)–Maritimes covering Eastern Canada, and British Columbia (BC)—Prairies covering Western Canada; (ii) Current recycling plants are located in the provinces of BC, ON, and QC as the base of operations [7,53]; and (iii) ON, QC–Maritimes, and BC and Prairies are geoeconomic clusters based on employment (number of jobs) in the electrical equipment manufacturing sector [54]. In the ON, QC–Maritimes, and BC–Prairies recycling clusters, 81, 63 and 107 collection sites, respectively, were filtered over a total of 97, 100 and 140 potential collection sites, respectively. The number of collection sites chosen in the QC–Maritimes cluster was the smallest in comparison with the other two clusters due to the low population density in the PCs located in the Atlantic Provinces (seven collection sites chosen over 25 candidates). The spent battery mass from each Canadian province is allocated to collection sites located in selected PCs using an additional criterion based on the number of PCs with a household annual income over CAD 100,000. In the ON cluster, the three major PCs with the higher battery mass allocated are Toronto, Hamilton, and Ottawa with 16, 16, and 5 collection sites, respectively. Figure 3 illustrates a sample of this analysis for the ON cluster.

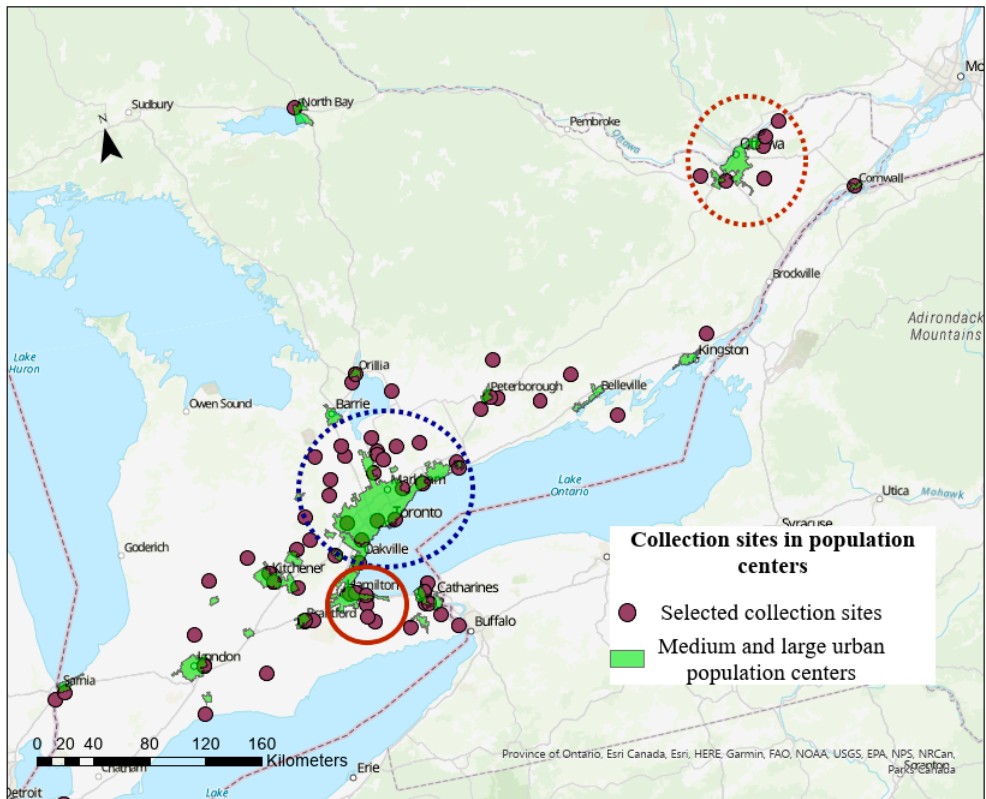

**Figure 3.** Analysis of a sample spatial analysis of suitable collection sites based on the siting criteria of near distance to large population areas, Ontario recycling cluster. The three major population centers with more collection sites and higher battery mas allocated are Toronto inside the dotted blue circle, Hamilton in the brown circle, and Ottawa inside the dotted red circle.

Moreover, the algorithms considered the battery mass allocation at the collection sites, which is why the optimal dismantling hub is located closer to the zone with the greatest number of scrapyards and the recycling facility is not halfway between the dismantling facilities chosen considering the criterion to be located closest to rail stations. In the end, recycling processing facilities can be located within or near high population centers and with high accessibility to a large quantity of spent LIBs from electric vehicles. The results from the location-allocation optimization of the dismantling hubs, recycling processing plants, and aluminum, copper, and steel smelting facilities are illustrated in Figure 4. The number of facilities (dismantling hubs and recycling processing plants) were iterative and estimated in order to minimize the truck transport distance between the demand points and the facilities chosen.

In the ON recycling cluster, the best locations for dismantling hubs are Toronto, Hamilton, Barrie, London, Oshawa, and Ottawa. The Toronto, Hamilton, and Barrie dismantling hubs represent 26%, 23% and 19%, respectively, of the total battery mass allocated in the ON cluster. The dismantling hub located in Toronto met the criteria to be selected as the centralized recycling processing facility in the Ontario recycling cluster. Because all available aluminum smelters are in Quebec, all the dismantling hubs located in Ontario feed the Alcoa Lte aluminum smelter located in Bécancour, QC. The dismantling hubs located in Oshawa, Barrie, and Toronto feed the steel smelter of Gerdau Whitby Steel Mill in Whitby; meanwhile, the dismantling hubs located in London and Hamilton feed the steel smelter of Stelco located in Hamilton, and the Ivaco Rolling Mills Ltd. steel smelter located in L'Original, ON is fed by the dismantling hub located in Ottawa. All six dismantling hubs feed the Glencore Canada Ltd. copper smelter, Horne Foundry, located in Rouyn-Noranda, QC.

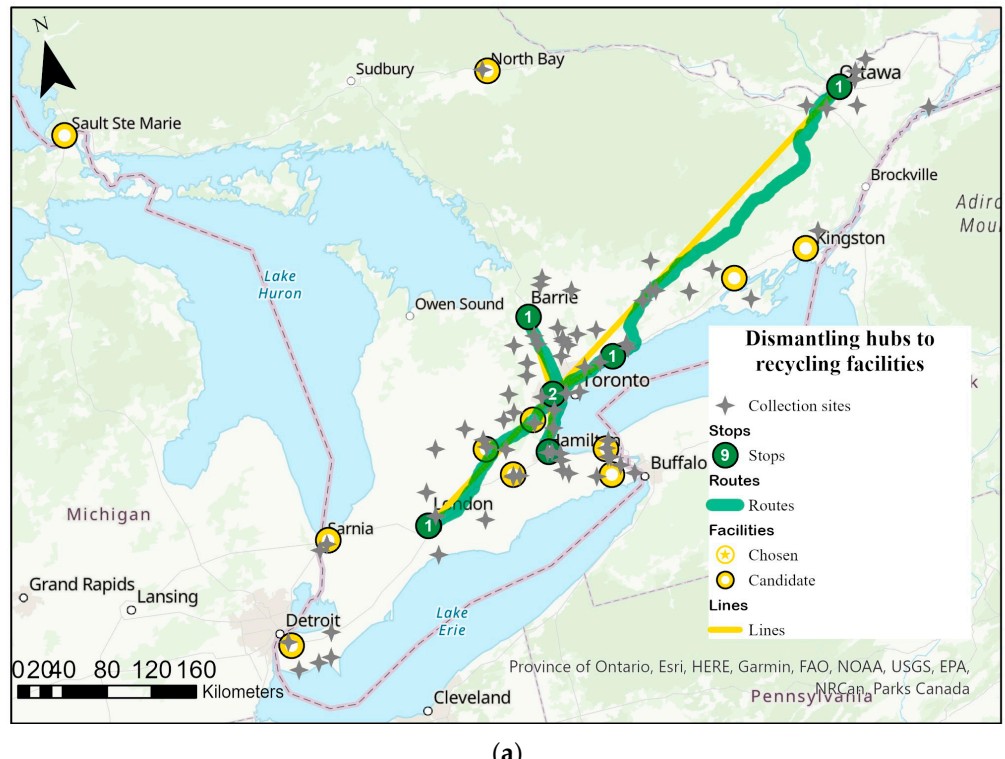

(**a**)

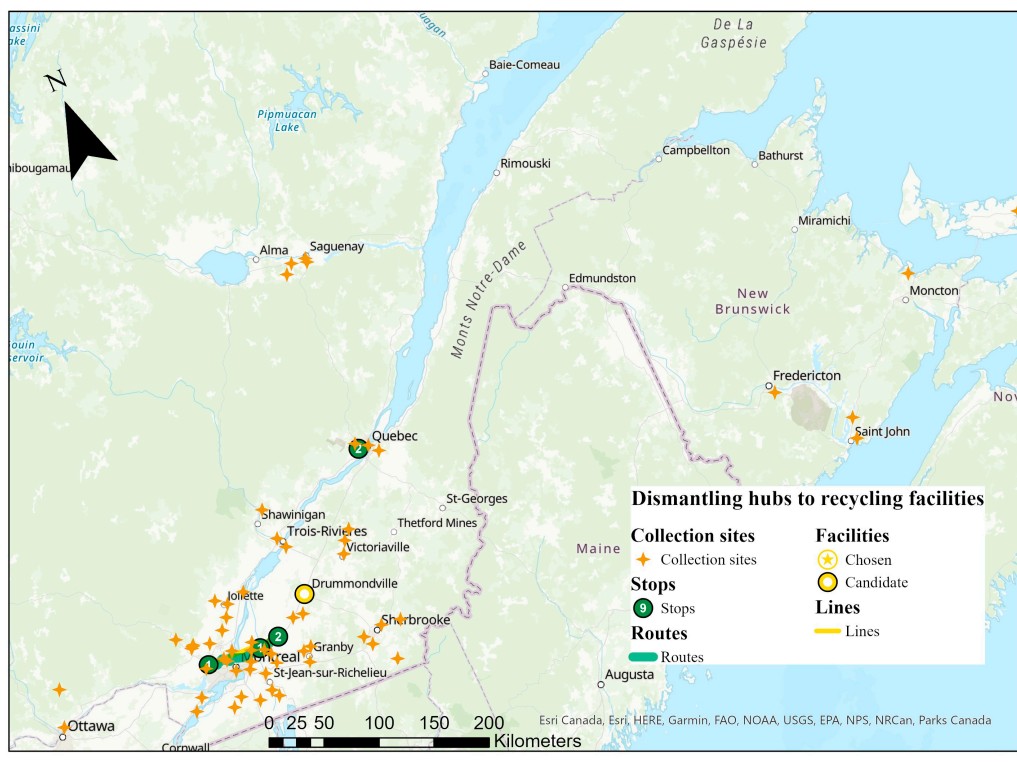

(**b**)

**Figure 4.** *Cont.*

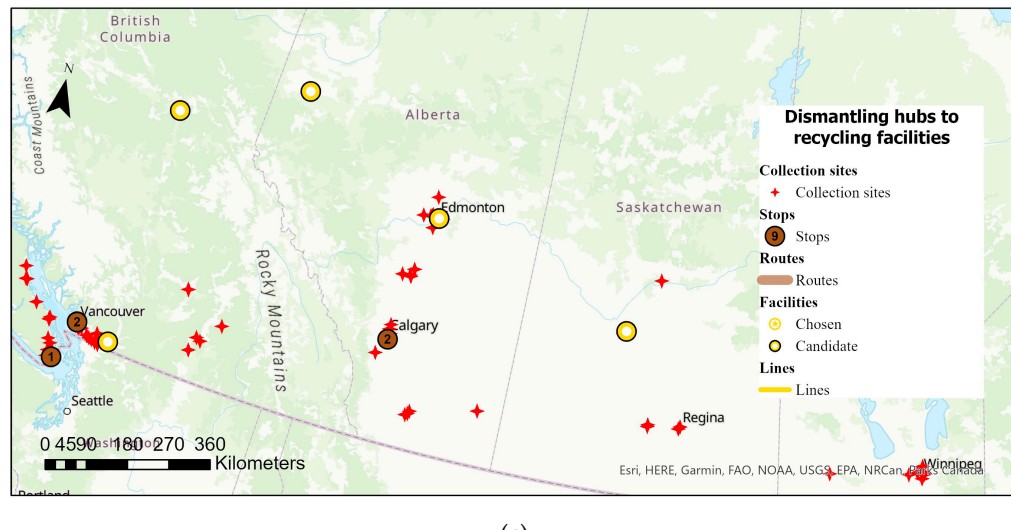

(**c**)

**Figure 4.** Geographical location of collection sites and optimal locations of dismantling hubs and recycling processing facilities. Transit routes from EV LIB dismantling hub to the closest recycling site using road transportation network. Dismantling hubs are identified as stop #1 (origin) and recycling processing facilities as stop #2 (destination), where the routes follow the direction toward recycling facilities. (**a**) Ontario recycling cluster; (**b**) Quebec–Maritimes recycling cluster; and (**c**) British Columbia–Prairies recycling cluster.

In the QC–Maritimes recycling cluster, two dismantling hubs located in Montreal and Beloeil feed the recycling facility in St. Hyacinthe, QC, and the dismantling hub located in Quebec, QC is also the same location to operate a recycling facility. The Montreal and Beloeil dismantling hubs represent 41% and 40%, respectively, of the total battery mass allocated in the QC–Maritimes cluster. These two dismantling hubs feed the Alcoa Lte aluminum smelter in Bécancour, QC, and the dismantling hub in Quebec feeds the aluminum Alcoa Lte aluminum smelter in Deschambault, QC. There were two steel smelters chosen; the ArcelorMittal Contrecœur, which is fed by the dismantling hub in Quebec, and the ArcelorMittal Montreal in St. Patrick, which is fed by the two dismantling hubs located in Montreal and Beloeil. All three dismantling hubs feed the Horne Foundry copper smelter located in Rouyn-Noranda, QC.

In the BC–Prairies recycling cluster, three dismantling hubs were chosen: Victoria and Vancouver in BC and Calgary in Alberta, where the dismantling hub in Vancouver represents 76% of the total allocation of battery mass in the BC–Prairies cluster. These dismantling hubs located in BC and AB also met the criteria to be chosen as recycling processing facilities, where the stops 1 (origin) and 2 (destination) have the same geolocation. There is only one aluminum smelter located in Kitimat, BC fed by all the dismantling hubs; similarly there is only one steel smelter in Edmonton, AB. The aluminum smelter belongs to Rio Tinto Alcan and the steel smelter to Alta Steel.

Transportation route optimization was performed for each road transportation network of the three recycling clusters to obtain accurate geodesic travel distances between origin (stop 1) and destination (stop 2) by finding the shortest path between stops. Figure 4 also illustrates the routes from the dismantling hubs to the recycling processing facilities.

### 3.2. Transportation Payload Distance and Environmental and Economic Impacts

The GIS optimization process is based on the minimization of the total ton–kilometers (t·km) transported between the collection sites and the EoL processing destination. The number of ton–kilometers is the weight in tons of battery pack/component transported multiplied by the number of kilometers driven. This is the appropriate objective function to minimize the economic and environmental costs corresponding to the total distance traveled by applying specific metrics and emission factors.

Currently, truck transportation is assumed as the primary mode for transporting spent batteries. Table S8 in the Supplementary Information presents the truck transportation payload distance of spent EV LIBs from demand points as either collection sites or dismantling hubs to EoL processing facilities (dismantling hubs and recycling and smelting facilities). The total ton–kilometers in the ON, QC–Maritimes, and BC–Prairies recycling clusters represents 19%, 38%, and 43%, respectively, of the total payload distance of the reverse logistics network. BC–Prairies has the longest distance travelled because of the distance from the dismantling facilities to the only aluminum smelter in the BC–Prairies cluster; meanwhile, QC–Maritimes has the second longest distance travelled because of the longest distance from all dismantling facilities to the only copper smelter in Canada.

The payload distance from the collection sites to the dismantling facilities varies among the three clusters. The ON cluster presents the shortest distance (12% of the total distance travelled to the dismantling facilities in all three clusters); meanwhile, QC–Maritimes and BC–Prairies represent 33% and 56%, respectively. In the Ontario scenario, there is a small volume and short distance because the dismantling hubs are located closer to scrapyard areas with more density; meanwhile, the QC–Maritimes cluster presents a long distance, and there are small-volume, out-of-province collection sites in the Maritimes. In the BC–Prairies cluster scenario, there is a long distance from the Prairies and a large volume in BC. This can be impacted by the number of dismantling facilities and differences in demand.

In the Ontario scenario, the truck transportation payload distance from the collection sites to the dismantling hubs and then to a recycling facility is 27% of total ton–kilometers transported within the ON cluster. The travelled distance from the dismantling facilities to the aluminum and copper smelters represents 73% of the total ton–kilometers transported within the ON cluster because there is out-of-province transportation from the dismantling hubs in ON to the smelter facilities in QC.

In the Quebec–Maritimes scenario, transportation from collection sites to the dismantling facilities and from dismantling hubs to the copper smelter facility represents 21% and 62%, respectively, of the total ton–kilometers transported within the QC–Maritimes cluster. Similarly, transportation from the collection sites to the dismantling facilities in the BC–Prairies cluster represents 31% of the total ton–kilometers in the out-of-province scenario in the West cluster, considering that the travelled distance from the dismantling facilities to the aluminum smelting facility represents 63% of the total ton–kilometers transported within the BC-Prairies cluster.

Aluminum, steel, and copper scrap smelter locations have a fixed effect in the model. This is reflected by the fact that most aluminum smelters are in QC. Therefore, the QC–Maritimes travel distance to aluminum smelters is shorter than the ON travel distance. At this point, increasing or decreasing the number of dismantling facilities depends on the demand of spent batteries, which can be controlled for the optimization process. For example, the longest ON travel distance to aluminum smelters could potentially be even further if there were more dismantling facilities in ON as consequence of an increase in available spent batteries. Currently, ON is not leading the EV sales in Canada despite being the most populated province. As we move toward 2035 (or 2050) where ICEs are to be removed and EV adoption is tied to population, LIB recycling demand will increase in ON.

The transportation payload distance has a direct influence on the environmental and economic impacts of implementing a recycling end-of-life EV LIB infrastructure.

Life-cycle GHG emission results for transporting 1 kg of the spent battery packs from the EV collection sites to the spent battery processing facilities are indicated in Table S9 in the Supplementary Information. This outcome shows the importance of understanding the environmental impacts of spent battery logistics at different points in the reverse logistics network when assessing the environmental sustainability of spent battery recycling for all three geospatial scenarios. The life-cycle GHG emissions for battery transportation along all the reverse logistics networks in the QC–Maritimes East cluster present the lowest environmental impact regardless of the long distance from the collection centers in the Maritimes because of its low tonnage. Meanwhile, the BC–Prairies West cluster ended up

with the highest GHG emissions due to a lack of infrastructure, and the ON East cluster ended up in the middle.

In Figure 5, the life-cycle environmental impact for transporting 1 kg of spent EV LIB in the ON, QC–Maritimes and BC–Prairies recycling clusters is 0.063, 0.045 and 0.081 kg $CO_{2e}$, respectively. The transportation emissions from the collection sites to the dismantling and recycling facilities represents 27%, 28% and 28% of the total emissions in ON, QC–Maritimes, and BC–Prairies, respectively; meanwhile, the emissions for the transportation of aluminum scrap from the dismantling hubs to a smelting facility represents 41% and 66%, respectively, of total emissions in ON and BC–Prairies.

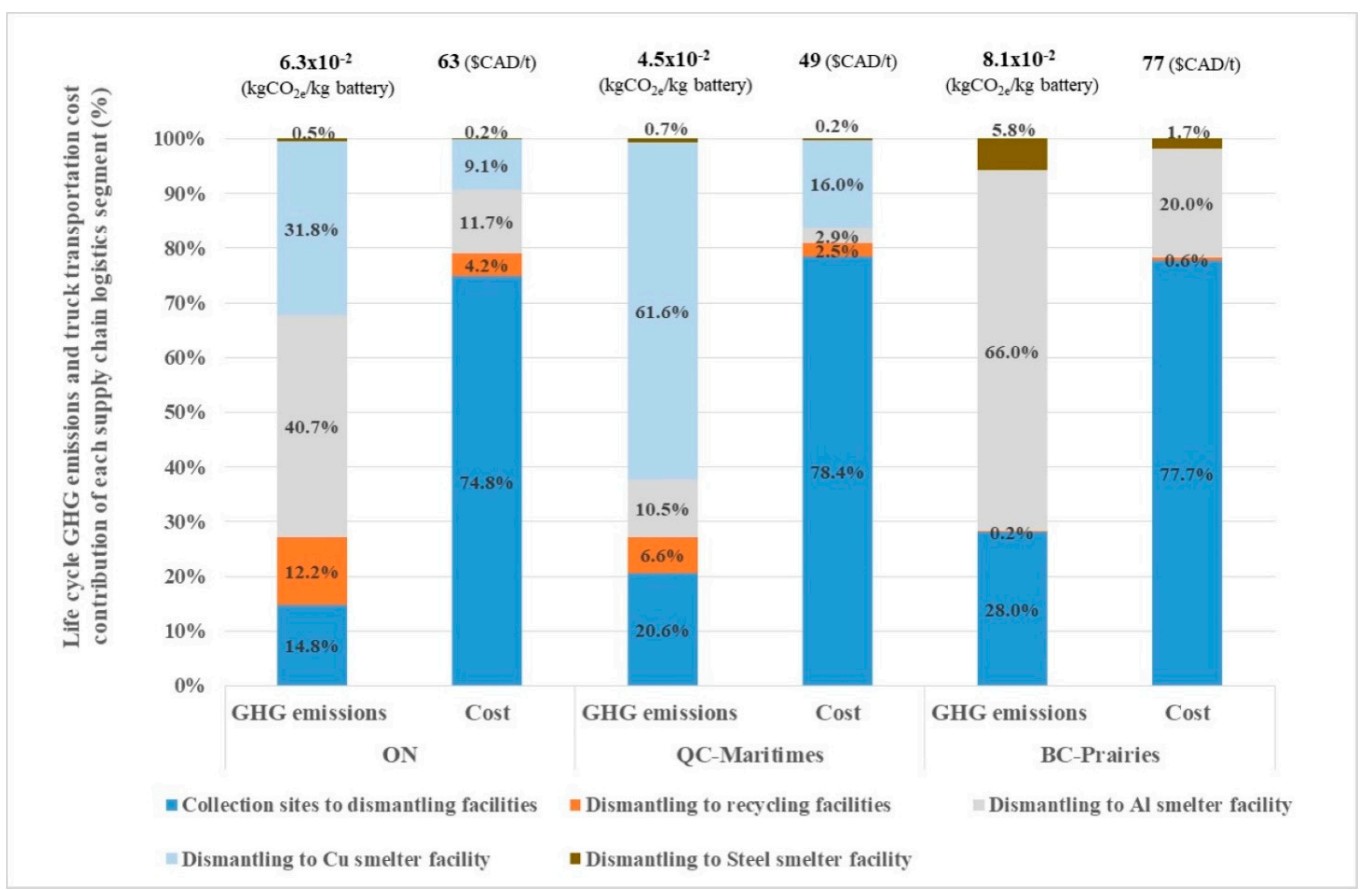

**Figure 5.** Overall GHG emissions (1 kg of spent EV LIB pack) and cost (CAD/t of spent EV LIB pack) impact of the overall logistics for three regional clusters. Contribution breakdown (%) is provided for each reverse logistics segment (Bottom).

In regards to the integrated life-cycle environmental impact of the spent battery pack at EoL, including transportation and recycling processing in the LCA system boundary, the total life-cycle GHG emissions for 1 kg of the spent battery pack recycling in the ON, QC–Maritimes, and BC–Prairies recycling clusters is estimated to be 1.17, 1.15 and 1.19 kg $CO_{2e}$/kg battery pack, respectively, where the emissions for transportation represents 5%, 4% and 7%, respectively, of the total emissions by battery pack recycling processes. The relative share of the environmental impact of only battery cathode material recovery, i.e., the total life-cycle GHG emissions of battery cathode materials recovered from recycling processing and emissions of transportation from collection sites to dismantling hubs and then to recycling facilities, accounts for 24% and 7%, respectively, of the total emissions of battery-grade cathode and battery pack production from virgin materials. The share of the life-cycle GHG emissions of the battery cathode of the total emissions of battery pack production from virgin materials accounts for 28%. The average estimated values of life-cycle GHG emissions of recycled cathode raw materials, battery cathode production from

virgin materials, and battery pack production from virgin materials (0.7, 2.93 and 10.4 kg $CO_{2e}$/kg of the spent battery pack, respectively), including transportation are indicated in Section S11 and Table S10 in the Supplementary Information.

The economic costs of the transportation network of the spent battery supply chain for recycling processes is calculated based on the estimated transportation unit cost for each of the segments of the reverse logistics network, i.e., collection sites to dismantling facilities and from the later locations to recycling/smelting facilities, considering Table S11 and the correspondents optimized payload distances in Table S8 in the Supplementary Information.

As a result, the truck transportation costs of 1 t of the spent battery packs from the EV collection sites to the battery processing facilities are indicated in Table S12 in the Supplementary Information. Taking into account only the transportation costs of the spent battery cells for recycling, i.e., truck transportation from collection sites to dismantling hubs and then to the centralized recycling facility for each geospatial scenario, these costs are CAD 50/t, CAD 40/t, and CAD 60/t for the ON, QC–Maritimes, and BC–Prairies clusters, respectively, which include the regular loading/unloading and distance-dependent travel costs, as well as the handling fee and other costs for transporting class 9 hazardous goods, such as LIBs. The average estimated transportation costs of the spent batteries to recycling facilities in Canada is CAD 50/t of the spent battery pack, which represents 4% of the total operating costs of battery recycling. Based on the study by Baxter [55], the operating costs of spent battery recycling processing is estimated at CAD 1244/t of the spent battery pack for a hydrometallurgical recycling facility in Canada. Hence, the average estimated cost of recycled cathode raw materials is CAD 1.29/kg of the spent battery pack, including transportation.

In Figure 5 showing the ON recycling cluster, truck transportation costs to the dismantling facilities represent 74.8% of the total truck transportation; meanwhile in QC–Maritimes and BC–Prairies, these costs contribute 78.4% and 77.7% of the total truck transportation, respectively. By adding the transportation costs of scrap metal recycling, i.e., truck transportation from the dismantling facilities to aluminum, copper, and steel smelting facilities for each geospatial scenario, the total truck transportation costs of the spent battery pack recycling increase by CAD 13.3/t, CAD 9.4/t, and CAD 16.6/t for the ON, QC–Maritimes, and BC–Prairies clusters, respectively. Note that all the aluminum smelting candidates in the East cluster are located in Quebec and the aluminum scrap from the dismantling facilities in Ontario is transported to the chosen aluminum smelter in QC, thereby increasing the payload distance. In the West cluster, however, there is only one aluminum smelter candidate in northern BC, and it is assumed, one steel smelter candidate in Alberta that increased the total payload distance. Importing copper to a closer smelter facility in the US was not evaluated. The QC–Maritimes cluster presents 29% and 44% less life-cycle carbon footprint than the ON and BC–Prairies clusters, respectively. The total truck transportation costs are 22% and 36% lower than ON and BC–Prairies clusters, respectively.

Overall, the truck transportation from the collection to dismantling facilities has a significant impact on the overall costs but much less on the GHG emissions. However, this conclusion may change if fuel prices increase due to the introduction of the carbon tax and higher crude oil prices.

### 3.3. Carbon Intensity of Transportation Routes

The optimized transportation routes from collection sites to dismantling hubs, dismantling hubs to recycling processing facilities, and dismantling hubs to smelters, were merged as a single route network using different geoprocessing tools (Merge, Buffer, Intersect, and Dissolve) of ArcGIS Pro© software. This helped with the generation of a linear heat map [56] as $kgCO_{2e}$/km, linking hot spots of GHG emissions with the most frequently traveled trucking routes. These linear heat maps for each recycling cluster are shown in Figure 6. Linear heat maps show high-intensity lines (hot spot) in red and low-intensity lines in yellow.

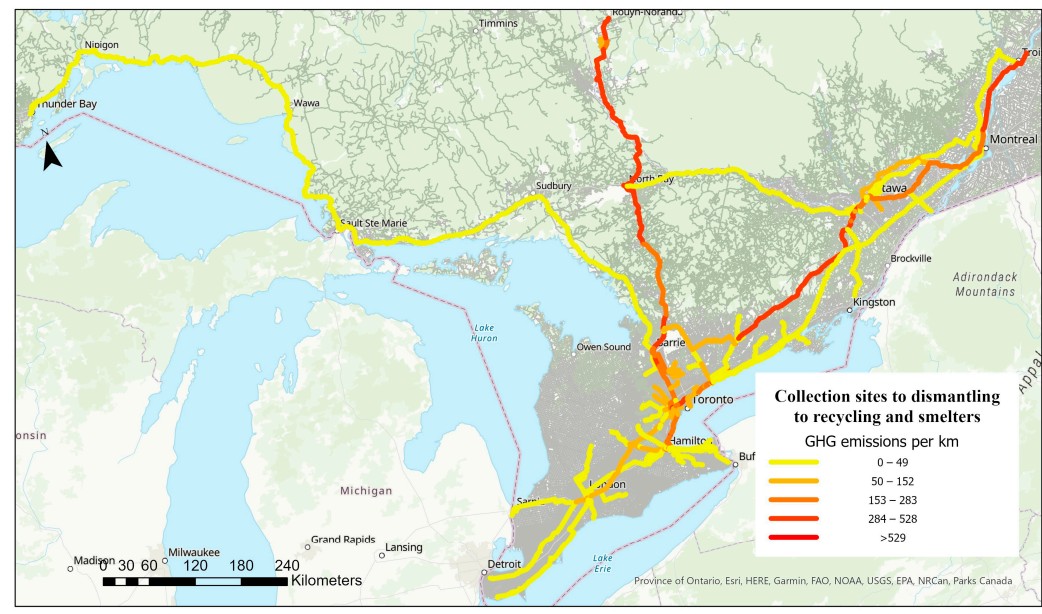

(**a**)

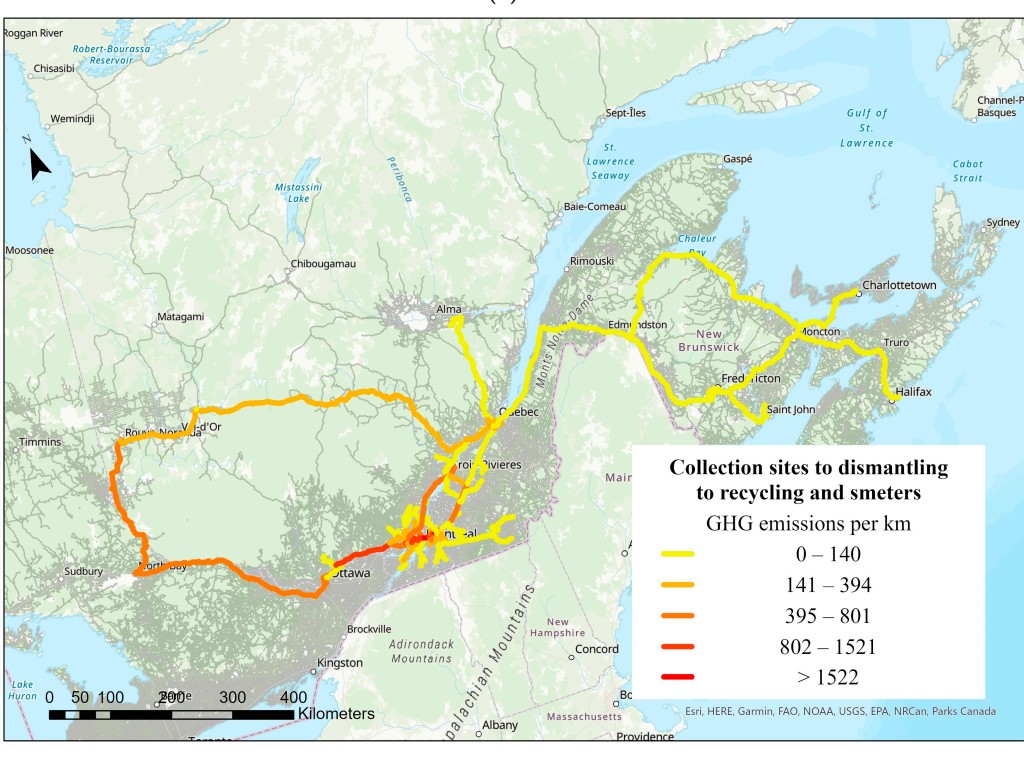

(**b**)

**Figure 6.** *Cont.*

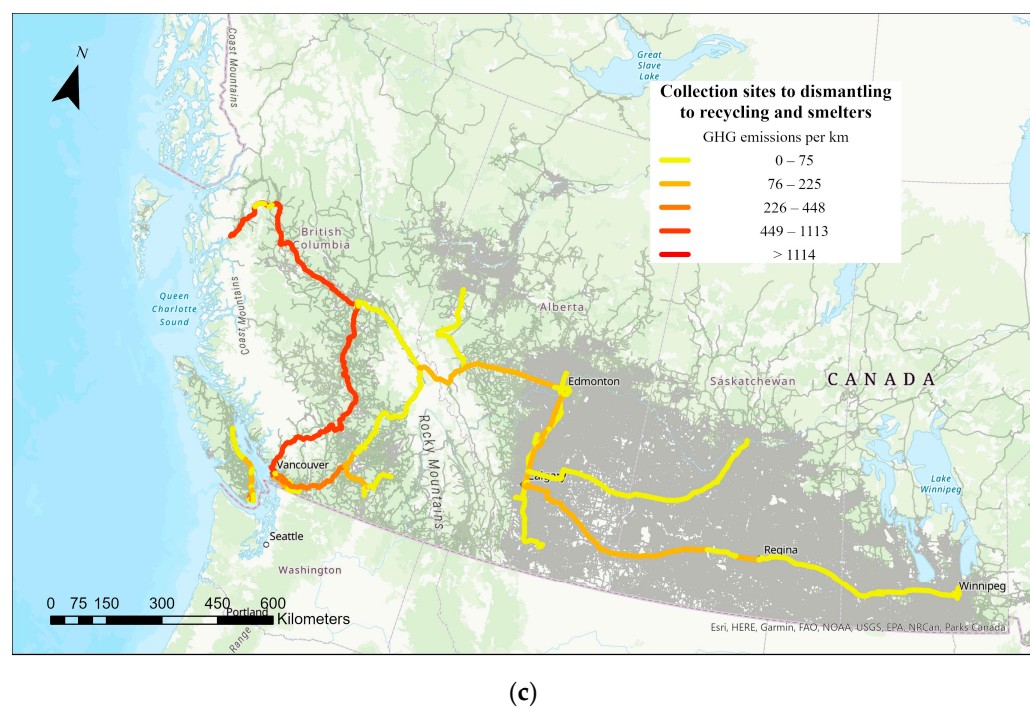

(**c**)

**Figure 6.** Linear heat map of carbon intensity of merged transportation routes of collection sites to dismantling hubs to recycling processing and smelting facilities for the Ontario, Quebec–Maritimes, and the BC–Prairies recycling clusters. (**a**) Ontario recycling cluster; (**b**) QC–Maritimes recycling cluster; and (**c**) BC–Prairies recycling cluster.

Overall, the low-intensity lines (in yellow) correspond to the routes of collection sites to dismantling hubs for all three recycling clusters. However, the emission intensity ranges vary among clusters where the QC–Maritimes cluster has the highest range up to 140 kg $CO_{2e}$/km and Ontario presents the lowest, with up to 49 kg $CO_{2e}$/km due to the scattered location of the collection sites in the QC–Maritimes cluster, a similar scenario to the BC–Prairies cluster. There are mild-intensity lines (in brown) that reach the location of the centralized recycling processing facility, i.e., Toronto (ON cluster) and St. Hyacinthe, (QC–Maritimes cluster) because of the concurrent routes from collection sites to dismantling hubs and from dismantling hubs to recycling processing facilities. In the QC–Maritimes cluster, there are mild-intensity lines from the dismantling hubs to the aluminum, copper, and steel smelters. In the BC–Prairies cluster, there are route segments with mild-intensity lines from the dismantling hubs in Vancouver, BC and Calgary, AB to the steel smelter in Edmonton, AB. The emissions intensities per km travelled are in the range of 153–283, 395–801, and 226–448 kg $CO_{2e}$/km. The high-intensity lines (in red) correspond to the routes of dismantling hubs to aluminum smelters and dismantling hubs to copper smelters for the ON cluster; the segment routes from the collection sites to the dismantling hub in Montreal integrated with the segment routes from the Montreal dismantling hub to the copper smelter; and the route segment from the dismantling hub in Vancouver, BC to the aluminum smelter in Kitimat in the BC–Prairies cluster.

### 3.4. Forecasted Recycling Processing Capacity

A baseline MFA scenario for spent EV LIBs was used in this study to forecast the annual number of LIBs available for recycling during the time period of 2022 to 2040. In this scenario, the number of EVs in Canada, specifically ZEVs, is expected to increase at a CAGR of 10%, and the number of LIBs for recycling in 2040 would be 130 times the potential current level (Table S2). Likewise, under a net-zero MFA scenario for the same period of time, which is described in Section 2.1.1, the quantity of EV registrations would grow at a CAGR of 25% to meet Canada's 2035 goal of reaching a 100% share of net-zero EVs over total

passenger vehicle sales (Table S3). Under the analysis of these two scenarios and regardless of the uncertainty inherent to the assumptions made to determine long-term forecasts, the results suggest a substantial increase in spent EV LIBs to be diverted to recycling processing facilities in Canada by 2040. The challenges of building an EV battery EoL infrastructure are even greater in terms of facility capacity, GHG emissions, and economic costs per kilometer. The annual recycling processing capacity of the ON, QC–Maritimes, and BC–Prairies recycling clusters estimated in the net-zero MFA scenario would increase by a factor of 4.5, 4.6 and 4.3, respectively, in comparison with the annual capacity estimated in the baseline MFA scenario. The GHG emissions and transportation costs per kilometer present the same number of increments per recycling cluster. Keeping the location of dismantling hubs and recycling processing facility candidates and the optimal locations of the collection sites simulated for the baseline MFA scenario, the allocation of the battery mass of the net-zero MFA scenario was used to calculate the optimal recycling processing capacity and payload transportation distances. The estimated annual recycling processing capacity under these two scenarios by 2040 is illustrated in Figure 7.

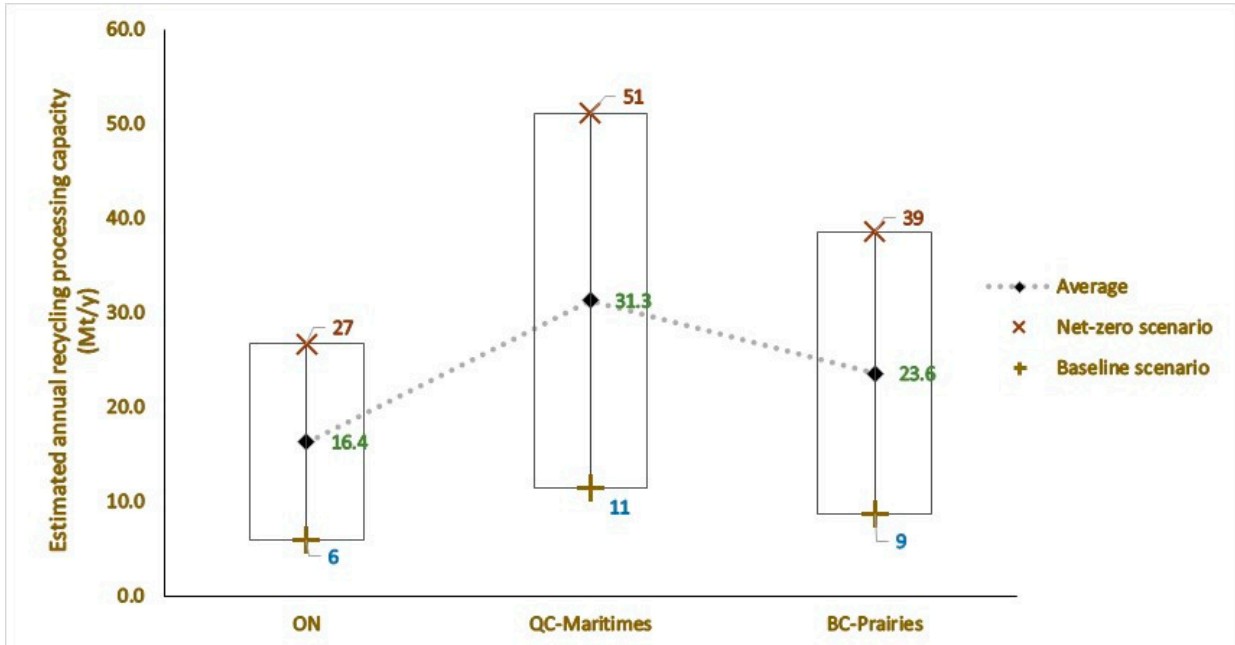

**Figure 7.** Estimated annual recycling processing facility capacity for the Ontario, Quebec–Maritimes, British Columbia–Prairies recycling clusters assumed to be viable by 2040 under the baseline and net-zero MFA scenarios. The lowest and highest potential recycling processing capacities presented in Ontario and QC–Maritimes clusters reflects the level of viability of local supply of spent EV LIBs in these provinces.

Recycling profitability can be achieved through economies of scale; for example, in a UK recycling facility, the profitability threshold could increase to >50,000 tons per year for pyrometallurgical and circa 17,000 tons per year for hydrometallurgical recycling [16]. In Figure 7, the baseline scenario presents low levels of economic viability for all the recycling clusters; hence, complementing this with an out-of-country recycling model would be an alternative to compare with the levels of viability of the in-country recycling model presented in the net-zero MFA scenario.

*3.5. Further Work*

The reverse logistics model of EV LIBs described in this work is flexible and can be adapted to integrate other jurisdictions in North America. The integration of US processing facilities closer to Canadian facilities could generate different optimal EoL management

locations and optimal transportation routes. The network routing optimization in this model could be enhanced by integrating an LIB tracing and tracking system to ensure safe and environmentally friendly EoL battery management. The development of block chain technology solutions is an opportunity to increase efficiency and transparency in the spent battery supply chain through securely monitoring and answering the questions where, when, and how a spent battery can be collected to be transported to an EoL management facility [57]. Addressing battery traceability issues avoids the increased risk of diverting spent batteries to landfills and recycles them instead. Nevertheless, policy frameworks and regulatory mechanisms need to be addressed in terms of the final destination of the batteries [29].

In the context of the circular economy, the recovery of critical and strategic metals, such as cobalt and nickel, to produce cathode LIBs in Canada could generate a more reliable and complete set of environmental and economic data to improve hotspot analyses for the recycling process of spent EV batteries [58]. A closed-loop approach can be applied by integrating this spatial model, and its findings relate to the environmental implications of transporting spent batteries to recycling facilities with a life cycle assessment of spent EV LIB recycling processing options and its environmental credits as a result of recovering battery cathode material to be reused in the production phase of LIBs. A regionalized LCA of specific geographical locations, such as the provinces of Ontario, Quebec, and British Columbia, would be performed to be consistent with the battery transportation GHG emissions calculated in this study.

## 4. Conclusions

This study presents a spatial modeling framework to quantify the environmental and economic effects of the expansion of the supporting infrastructure network for EV LIB end-of-life management in Canada, based on the integration of geographic information system, material flow analysis, life cycle assessment (truck transportation emission intensity), and lithium-ion battery truck transportation costs. Although battery recycling processing options will be critical to diverting EV batteries from EV waste streams, the overall impacts of the reverse logistics (collecting, dismantling, and recycling) of these batteries must be considered. Because the collection and transportation of electric vehicle batteries have an important contribution on the environmental and economic overloads of the end-of-life infrastructure, there is a need for an optimal management and responsive recovery at end-of-life logistics.

The main conclusions of this case study are the following:

The allocation of battery mass at the collection sites, the road network travel distance minimization, and the facility placement criteria are some of the most important parameters that define the optimal locations of the dismantling hubs and recycling processing facilities and enable the analysis of the environmental and economic impacts on the end-of-life infrastructure network for EV LIBs.

The assumptions considered in the model design criteria have impacted the results of this study. Spent battery mass availability assumptions would change with EV market share projections, evolving battery chemistries, and used battery imports from other provinces or countries outside Canada. Using different ZEV adoption policies and government incentives could also adjust the total battery mass generated and the spatial distribution, thus affecting the optimal facility locations, transportation distances, and overall environmental and economic impacts.

In the three regional clusters for the collection and transportation of spent EV batteries in Canada, the transportation network optimization provided the best solution to allocate spent batteries to dismantling facilities and to locate the closest recycling/smelting facilities based on the payload distance parameter.

Life-cycle carbon footprints and transportation costs were calculated for all the optimized routes along the spent battery supply chain for recycling processes in Canada. The recycling cluster in Quebec presents the lowest life-cycle GHG emissions and transporta-

tion unit costs in comparison to the two recycling clusters located in Ontario and British Columbia due to Quebec having the largest share in EV registrations, which is largely influenced by government incentives.

The overall life-cycle GHG emissions of the spent battery pack recycling was obtained by adding life-cycle GHG emissions from transportation to emissions resulting from battery cell recycling processing and other metal recovery. In terms of the relative share of the environmental impacts of only battery cathode recycling, GHG emissions for transportation account for 2% to 3% of total life-cycle GHG emissions of battery cathode recycling. Furthermore, the latter represents 7% of the total life-cycle GHG emission of battery pack production from virgin materials that is lower than the 28% of the environmental impact share of the battery cathode production from virgin materials.

Truck transportation from the collection to the dismantling facilities had a significant impact on the overall costs but much less on the GHG emissions. There was a differentiation of transportation costs between the collection sites to the dismantling facility segments in comparison with the other segments in the transportation network due to the added costs for the complexity of hazardous material transportation of spent LIB packs. The same GHG emission factor was applied to each segment Thus, the proposed optimal facility siting that minimizes transport distance to dismantling hubs was suggested to reduce the overall costs of transportation, ensure safety compliance, and facilitate the feasibility of building a recycling facility. However, it is also important to take into account the regional regulatory framework related to the operational transportation costs that may lead to set up place-specific parameters.

**Supplementary Materials:** The following supporting information can be downloaded at: https://www.mdpi.com/article/10.3390/su142215321/s1. Supplementary data: EV demand forecast tables, geospatial data inputs, estimated truck transportation distances, GHG emissions and costs tables. References [59–62] are cited in the supplementary materials.

**Author Contributions:** Conceptualization, G.G.-C., B.Y. and F.B.; Data curation, G.G.-C.; Formal analysis, G.G.-C.; Investigation, G.G.-C.; Methodology, G.G.-C.; Project administration, B.Y.; Supervision, B.Y. and F.B.; Validation, G.G.-C.; Writing—original draft, G.G.-C.; Writing—review and editing, G.G.-C., B.Y. and F.B. All authors have read and agreed to the published version of the manuscript.

**Funding:** This research received no external funding.

**Institutional Review Board Statement:** Not applicable.

**Informed Consent Statement:** Not applicable.

**Data Availability Statement:** Data supporting reported results in Supplementary Information File.

**Acknowledgments:** The authors gratefully acknowledge financial support from the Office of Energy Research and Development (OERD) of the Natural Resources Canada (project number NRC-19-109) and the Advanced Clean Energy (ACE) program of the National Research Council of Canada.

**Conflicts of Interest:** The authors declare no conflict of interest.

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
