# Peer review of "Development of a Reverse Logistics Modeling for End-of-Life Lithium-Ion Batteries and Its Impact on Recycling Viability—A Case Study to Support End-of-Life Electric Vehicle Battery Strategy in Canada"

_sustainability, doi:10.3390/su142215321_

Round 1

Reviewer 1 Report

Congratulations on such an insightful manuscript that addresses some of the world's current environmental concerns.

The manuscript is well written, but it could benefit from a more thorough review of literature and the inclusion of more sources.

Reviewer 2 Report

The author(s) in this research developed a reverse logistic network for end-of-life Lithum-Ion Batteries and their impact on recycling viability.  The research is good.  Some of my suggestions for improving the quality of the paper are:

1. Section 1: Introduction: The author(s) quickly rush through the topic.  This part needs more detail with respect to the reverse logistics in general.  Authors also need to discuss more detail with respect to lithium ion batteries and their relationship to reverse logistics operations.

2.  Section 2:  Literature review needs to be added to enhance the quality of paper.  It can also detail the spatial modeling papers related to reverse logistics. 

3.  There are a number of spatial modeling data with respect to reverse logistics operations.   Authors need to explain on what basis the spatial modeling discussed in sections 2.1.1, 2.1.2, and 2.1.3 have been shortlisted and analysis done in this research.  

4.  Section 3, page 9, Line 268-270.  More details on how the recycling clusters were selected need to be given.

5.  In addition to figures 4 and 5 , give salient data of them should be given on separate tables for clarity.

6.  Results of the study indicate that truck transportation from collection to the dismantling facility has an impact on overall cost and less on GHG emissions.  Reasons for this may be explained in the conclusion part of the paper.  

7.  

7.  

Reviewer 3 Report

Research on modeling a reverse logistics framework does not contribute significantly to the literature considering the subjects of logistics and sustainability. However, the choice of the lithium-ion battery as the study object enhances the proposal, complementing its innovative content. In addition, the combined analysis with GIS, MFA, and LCA tools enriches the research proposal, making it an interesting publication in the Sustainability journal.

The work presents an adequate conceptual basis. The introductory section of the work must associate the conceptual basis with the proposal to identify the research gap in the research explicitly. To this end, the authors must create a table specifying the most relevant articles in the area and how the manuscript complements the state-of-the-art with its research proposal.

The work presents a high potential for application in LIBs reverse logistics when considering Canadian facilities and the application of tools available to professionals involved in similar analyses.

The research methodology is adequate to achieve the research objectives. GIS, MFA, and LCA generated a robust approach to achieving the research objectives.

The discussion of the results is comprehensive enough to verify the research objectives' achievement. The debate on aspects considering the existing Canadian collection and recycling system promotes the enrichment of the research. The advance in research through the proposal identified in the future works section allows for directing researchers in related investigations.

The elements selected in conclusion represent an adequate synthesis of the results obtained in the research.

The quality of the text is adequate to the level required for a publication in a high-impact journal like Sustainability.

Reviewer 4 Report

No reviews, analysis, results and discussion related to the impact on recycling viability that stated by the title of manuscript.

Lack of comprehensive literature review and mapping of the proposed research result(s), no statement about research novelty and proposed new knowledge or new findings.

Most of the titles of figures are sentence(s), explanation for the figure(s), not title(s).
